# Construction and Demolition Waste Ballast as a Pozzolanic Addition in Binary Cements: Characterization and Thermodynamic Stability

**Santiago Yagüe-García [1] and Rosario García-Giménez [2,*]**

[1]   Grado en Ingeniería Electrónica Industrial y Automática, Avenida de la Universidad, Villanueva de la Cañada, 28691 Madrid, Spain; syagugar@uax.es

[2]   Departamento de Geología y Geoquímica, Facultad de Ciencias, Universidad Autónoma de Madrid, 28049 Madrid, Spain

*    Correspondence: rosario.garcia@uam.es

**Abstract:** The use of raw materials from the recovery of construction waste is frequent. In this study, the waste is obtained from degraded railway ballast, made up of siliceous minerals. This material is added directly to the mixtures to replace part of the cement, forming a good quality cementitious product. The pozzolanic reaction is studied within the waste/lime system in a solid phase and water system for one year. Reaction products such as stratlingite, layered double hydroxide-type compounds, hydrated tetracalcic aluminate, and gels appear. The formation of stratlingite and gels was favored throughout the reaction. The reaction was monitored, calculating the saturation index.

**Keywords:** CDW ballast; pozzolanic reaction; geochemical simulation; fields of stability; stratlingite

## 1. Introduction

The ballast used as a railway track material wears out over time, forcing it to be removed from the track when it is in inadequate condition, becoming waste or being recovered for use in the lifecycle of other materials. A series of changes affect the fragments that make up the ballast throughout its useful life, such as the deterioration of its angular shapes and the appearance of sphericity, which negatively affect the cohesion of the track bed. Therefore, planned maintenance cycles [1] and the systematic study of particle rounding are necessary to control the mosaic effect of piled gravel [2,3]. Worn ballast, generated in large volumes, becomes residual material that can be used as cementitious material for the manufacture of eco-efficient green cements, reducing the environmental footprint of cement industries [4–6].

Mineralogically, ballast waste is formed by silicates and clay minerals [7]. Most are applied as inert materials or dumped in waste dumps or landfills, which poses significant problems from a technical, economic, and environmental point of view [8,9]. Crushing construction waste into fine recycled fine powder that is used as alternative binding agents provides a new approach to recycle these waste products. The preparation of recycled fine powder for application in new cementitious materials not only reduces the quantities of construction waste, but also involves precise dosing. In the case of CDW ballast, the material is already prepared. The storage of track–ballast waste in landfills continues to cause serious environmental and socio-economic problems [10]. However, despite the novelty of the preliminary works for its incorporation into the life cycle of the materials, an in-depth study of the kinetics of the pozzolanic reaction, the evolution of the hydrated phases, and their stability is necessary—a step fundamental to establish the scientific bases for the future use of these waste products as pozzolans [5,6].

Simulation and modeling tools are a powerful extension of traditional experimental investigations. Experimental studies and their uncertainties and weaknesses can be

discussed, and quantitative predictions can be advanced [11]. It is for this reason that geochemical calculation models that rapidly represent processes that extend over time were used in this work.

In the present work, the pozzolanic reaction of binary cements with addition of ballast waste is studied, evaluating the stability of the hydrated phases based on pozzolanicity measurements while monitoring the parameters of the aqueous solutions generated in the pozzolanic reaction. The saturation indices are related to the mineral stability fields where the phases formed during the pozzolanic reaction are stabilized throughout the considered time (in this case, one year). The thermodynamic analysis and stability of the pozzolanic reactions pointed to possible dissolution/precipitation processes in the experimental phases, which were observed as reaction products. A simulation is carried out using a geochemical calculation program (PHREEQC method) to consider the phases present in the pozzolanic reaction and which will be compared with the phases obtained experimentally in this study.

## 2. Materials and Methods

### 2.1. Materials

Two types of sample materials from quarries known to supply ballast to the railway track in rehabilitation on the PB Río Duero—Valladolid-Campo Grande Station section of the Madrid–Segovia–Valladolid High-Speed Line were used, as follows:

(a) The BW1 sample comprises hornfels or cornubianite rock with a granoblastic texture, although, in many examples, the regional schistosity can be seen very clearly. Hornfels are dark rocks with opaque colors and matte sheens. The storage mound was located at the aggregate manufacturing plant of Canteras y Construcciones S.A. (CYCASA), in Avila (Spain). (b) The BW2 sample from Canteras Cuadrado, located 4 km from Avila (Spain) was quarried from a dark-colored porphyry quarry face. It is a quartz monzonite commonly called porphyry. From the same family as the BW1 sample, it was selected with the aims of differentiating this material from the previous one (BW2), given its darker coloration. After analyzing the rock, the loss due to abrasion of gravel, the material to be processed [12,13], was studied using the Los Angeles abrasion test [14]. To accept/reject the ballast, the Los Angeles abrasion test was employed, which loses its angular dimensions and size in the test, which is considered an accelerated test of deterioration and wear of the railway track ballast.

The study of grain size in the preparation of the mortars containing ballast waste is essential; thus, Ostrowski et al. [15] considered that a substitution of 7% cement with ballast waste improved both mechanical and elastic properties when different granulometric sizes were combined, avoiding homogeneity. Hence, the granulometry of the material was studied with the Los Angeles abrasion test, indicating that all the samples can be considered as possible pozzolans, depending on the particle size [16].

Ordinary Portland Cement (OPC) is a cement type CEM I 52.5 R, supplied by the Lafarge Holcim cement company (Toledo, Spain). This type of cement, employed for the preparation of the different mixtures, is characterized by a clinker composition of between 95% and 100% of mass and between 0% and 5% of minor components (especially selected inorganic components that are used in proportions below 5% of mass, with respect to the sum of all the main and minor components) [17].

The cement particles were smaller than 63 μm in size and 47.41% of them passed through a 12 μm sieve mesh. The chemical and mineralogical compositions of OPC are presented in Table 1. The blended cements were prepared in a high-speed powder mixer to ensure homogeneity.

**Table 1.** Chemical composition of OPC (LOI = loss on ignition).

| Oxide | $SiO_2$ | $Al_2O_3$ | $Fe_2O_3$ | CaO | $Na_2O$ | $K_2O$ | $TiO_2$ | $P_2O_5$ | $SO_3$ | LOI |
|-------|---------|-----------|-----------|-----|---------|--------|---------|----------|--------|-----|
| (%) | 20.16 | 4.36 | 2.52 | 63.41 | 0.35 | 0.91 | 0.21 | 0.14 | 3.57 | 1.9 |

### 2.2. Methods

Chemical characterization of phases was carried out with X-ray fluorescence (XRF) using a Philips PW 1404 (Phillips, Madrid, Spain) and a Sc-Mo X-ray tube. Fineness was analyzed with laser-ray diffraction (LRD) using a Sympatec Helos 12 KA spectrometer (Helos, Clausthal-Zellerfeld, Germany) and isopropyl alcohol as the non-reactive liquid.

Sample mineralogical compositions were determined with random powder X-ray diffraction (XRD) on a Siemens D-5000 X-ray diffractometer (Siemenes, Berlin Germany) fitted with a Cu anode. Their operating conditions were 30 mA and 40 kV, with divergence and reception slits of 2 and 0.6 mm, respectively. The samples were scanned in (2 y) 0.041 steps with 3 s counts per step. The characterization and semi-quantification of the samples were performed using the random powder method operating from $3°$ to $65°$ $2\theta$ at a rate of $2°$/min with an internal rutile standard to qualify the amorphous phase. The measured standards were analyzed using the Match v.3 and Rietveld Full Prof. 98 Software with the Inorganic Crystal Structure Database (ICSD) and the Crystallography Open Database (COD).

SEM/EDX morphological observations and sample microanalysis were performed with an Inspect FEI Company Electron Microscope ((Hillsboro, OR, USA) equipped with an energy-dispersive X-ray analyzer (W source, DX4i analyzer, and Si/Li detector). The chemical composition represented the average value of ten analyses of each sample, and the values were presented alongside their standard deviations. These semi-quantitative analyses were performed on clean surfaces to prevent all sources of contamination, such as high sulfur concentrations, which might otherwise interfere with the EDX analysis of the mineral formula. The results were expressed in oxides (wt.%) and adjusted to 100%.

Pozzolanic activity method: pozzolanic behavior in a pozzolan/calcium hydroxide (lime) system was studied using the solid residues after applying an accelerated chemical method. After each period of 1, 3, 7, 28, 90, 180, and 360 days of the reaction, the sample was washed with acetone and dried in an electric oven at $60\ °C$ for 24 h, ending the pozzolanic reaction. The content of fixed lime was calculated as the difference between the CaO concentration (mmol/L) in the original saturated lime solution (17.68 mmol/L) and the content of that same compound in the solution at each set point in time. Extra-pure calcium hydroxide (Ph Eur, USP, BP) was used as the chemical reagent.

An aqueous solution was used to measure the concentration of aqueous species of interest for the simulation (Na, K, Mg, Ca, Al, and Si). These concentrations were determined using inductively coupled plasma mass spectrometry (ICP-MS) with an Elan 6000 Perkin–Elmer Sciex analyzer (Perkin Elmer, Seville, Spain) with an automatic AS91 injector.

Thermodynamic method: in this study, the evolution of the hydrated phases that formed was evaluated with PHREEQC geochemical software version 2.18 [18], which processed the data on the aqueous species concentrations of the sample products at different reaction times (1, 7, 28, 90, 180, and 360 days). PHREEQC geochemical code was used to optimize the pH values, due to the high uncertainties of experimental analyses. The Thermoddem database was used for the simulation [19] because all the minerals presented in this study are listed on the PHREEQC database. A positive saturation index (SI) indicates the tendency of a mineral to precipitate (oversaturated aqueous solution with regard to the mineral), whereas a negative SI value indicates the tendency of a mineral to dissolve (unsaturated aqueous solution with regard to the minimum level of saturation).

The initial data employed to perform the speciation/solubility calculations were pH and total analytical concentrations obtained with ICP/MS at $25\ °C$ of the dissolved species: $Na^+$, $Mg^{2+}$, $AlO^{2-}$, $K^+$, $Ca^{2+}$, $SiO_4^{2-}$, and $CO_3^{2-}$ of the pozzolanic assay (liquid phase). The mineral compositions selected for study were layered double hydroxide (LDH)-type compounds, stratlingite, Ca/Si gel = 0.8, Ca/Si gel = 1.2, and Ca/Si = 1.6.

The activities of the aqueous species were calculated during the simulation. The concentrations of aqueous species, measured in solution at different times, were introduced in the model at pH = 12. The aqueous carbonate (not experimentally determined in the solution) was equilibrated with calcite in the model.

## 3. Results and Discussion

### 3.1. Ordinary Portland Cement (OPC)

The chemical analysis of OPC, provided in Table 1, details a mineralogical composition of $C_2S = 8.95\%$, $C_3S = 64.75\%$, $C_3A = 7.29\%$, and $C_4AF = 7.65\%$.

### 3.2. Rock Samples

The XRD mineralogical analysis of BW1 and BW2 samples is provided in Table 2.

**Table 2.** Rietveld quantification of the initial BW1 and BW2 samples ($R_B$ and $\chi^2$ = agreement factors).

| Sample | Biotite (%) | Quartz (%) | Orthoclase (%) | Albite (%) | Labradorite (%) | Kaolinite (%) | Hematite (%) | Amorphous Material (%) | $R_B$ | $\chi^2$ |
|---|---|---|---|---|---|---|---|---|---|---|
| BW1 | 30 | 29 | 6 | 18 | n.d. | 7 | 4 | 6 | 16.5 | 3.3 |
| BW2 | 35 | 21 | 5 | n.d. | 23 | 6 | 4 | 6 | 12.3 | 3.5 |

Silicates are the most abundant minerals in the composition, with biotite predominating at 10.08 Å (8.76 2θ), 4.48 Å (19.80 2θ), 3.66 Å (24.28 2θ), 3.35 Å (26.58 2θ), 2.56 Å (35.02 2θ), and 2.45 Å (36.64 2θ); labradorite at 3.19 Å (27.96 2θ)–3.21 Å (27.78 2θ); and quartz localized at 10.08 Å (8.76 2θ), 4.48 Å (19.80 2θ), 3.66 Å (24.28 2θ), 3.35 Å (26.58 2θ), 2.56 Å (35.02 2θ), and 2.45 Å (36.64 2θ); followed by orthoclase at 13.25 Å (27.44 2θ). Kaolinite is evident in small proportions (7.16 Å = 12.44 2θ) and hematite at 2.69 Å (33.30 2θ). The difference between one sample and the other was the presence or absence of albite and labradorite (with darker tones). The compositions of the other minerals are all very similar.

In view of the chemical analysis (Table 3), it can be said that the sum of silica, aluminum, and iron oxides exceeded 85% with high concentrations of sodium and calcium for the BW2 sample containing labradorite. Its loss on calcination in this sample was slightly higher than in the BW1 sample. As a minority and notable element, we must mention Sr (189 ppm).

**Table 3.** Chemical analysis of the main oxides in the samples under study.

| Oxides (%) | BW2 | BW1 |
|---|---|---|
| $Na_2O$ | 4.78 | 2.97 |
| $MgO$ | 0.95 | 1.72 |
| $Al_2O_3$ | 19.60 | 13.79 |
| $K_2O$ | 4.06 | 2.85 |
| $CaO$ | 3.26 | 2.77 |
| $MnO$ | 0.03 | 0.11 |
| $TiO_2$ | 0.37 | 0.80 |
| $Fe_2O_3$ | 3.49 | 6.72 |
| $SiO_2$ | 63.49 | 68.27 |
| LOI | 1.22 | 0.67 |

The results of the pozzolanicity test are presented in Figure 1. The measures of fixed lime in the samples have a logarithmic trend, with growth being visible until day 90 of the reaction, the age at which the maximum values were reached before their subsequent stabilization. The thermally activated samples had a very slow lime fixation capability at the early stages, increasing over time—a capability that was still present at the end of one year in this study (360 days).

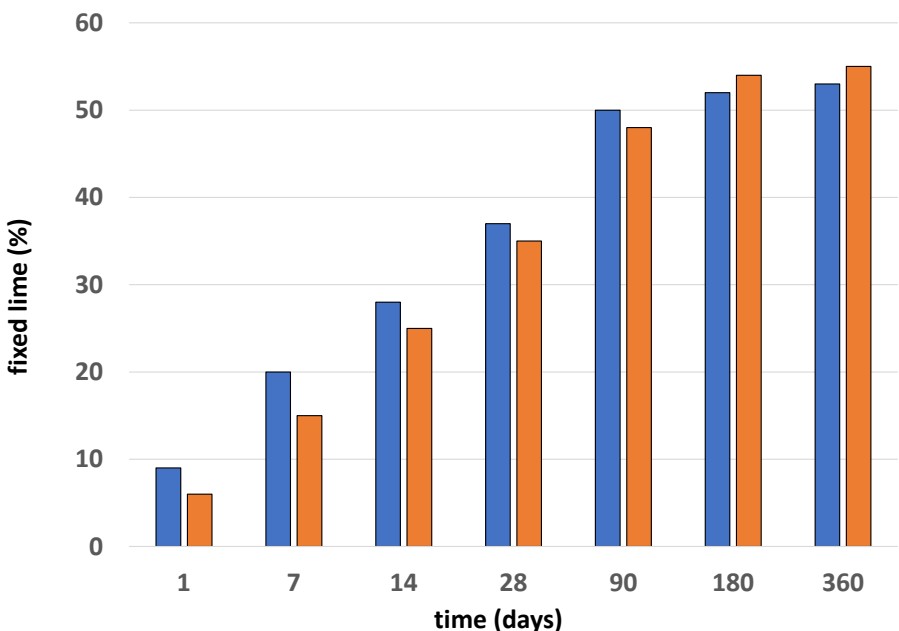

**Figure 1.** Pozzolanicity values of BW1 (blue) and BW2 (red) samples.

The results are consistent with those found in other investigations [20–22] "for similar materials". The ballast waste material will not need any type of activation because, as seen in previous research [16], the improvement in pozzolanicity with thermal activation is minimal. The thermal inactivation of the waste will favor the usual behavior of the C-S-H gels since throughout the experiments that were carried out with materials cured between 5 °C and 60 °C, the structural chains of the aforementioned C-S-H gels were polymerizing, giving rise to more porous and thick structures in the cement, which explained the low resistance at high temperatures [23].

### 3.3. Pozzolanic Reaction in the Cement Ballast Waste System

#### 3.3.1. Solid Phase

The kinetics of the pozzolanic reactions were observed by studying the hydrated phases that appeared in a solid state and that determined the acceptance or rejection of the reused waste (ballast waste) as a binder aggregate in the composition of OPC concrete.

The main studies on the identification of phases were carried out with X-ray diffraction analysis. Using this technique, the neoformed compounds of the saturated lime solution involved in the pozzolanic reaction within the waste and the calcium hydroxide were identified at different reaction times (1, 7, 28, 90, 180, and 360 days) (Figure 2).

The compounds identified in the ICDD PDF-4 information sheets were LDH-type compounds (04-015-1684), calcium aluminate hydrate ($C_3AH_{13}$), calcite (00-005-0586), stratlingite (01-085-8414), and C-S-H gels of low crystallinity, which are quantified in Table 4, as well as the initial minerals of the sample: quartz, feldspars, and phyllosilicates.

LDH-type compounds were cited in the literature as carboaluminates but, in recent times, they have considered as double oxides formed by super-imposed tetrahedral layers of lamellae with oxygen and an octahedral layer of aluminum with $(OH)^-$ group-type kaolinite (phyllosilicate 1:1) after undergoing dehydroxylation by calcination and a subsequent reaction with $Ca(OH)_2$ (placing the carbonate groups together with the hydroxyl groups) within the interlaminar spaces [22,24].

At the same time, C-S-H gels are recognized as compounds with a somewhat disorganized structure although they are arranged in nanometric-sized sheets, which are organized in chains of a finite number of elements. Since their nature is not very crystalline, they will be quantified as an amorphous material (Table 4).

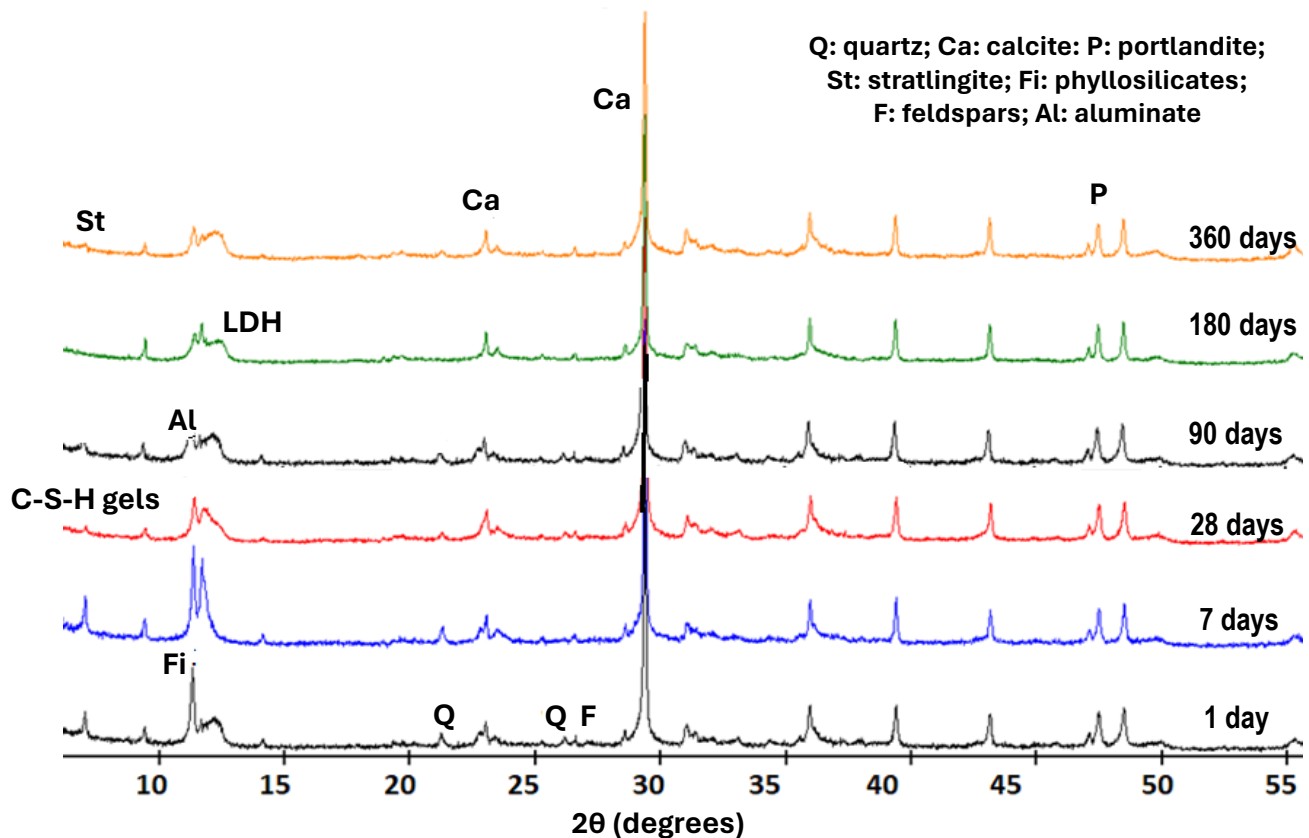

**Figure 2.** XRD patterns of the solid phases from ballast waste/lime system.

**Table 4.** Rietveld quantification of the solid samples from the ballast/lime system at different curing ages from BW1 and BW2 samples ($R_B$ and $\chi^2$, agreement factors, AM = amorphous material, Ca = calcite, F = feldspars, Fi = phyllosilicates, P = portlandite, Q = quartz, St = stratlingite, n.d. = not detected).

| Sample | Reaction Time (Days) | Q (%) | F (%) | Fi (%) | LDH (%) | St (%) | $C_3AH_{13}$ (%) | P (%) | Ca (%) | AM (%) | $\chi^2$ | $R_B$ |
|--------|------|----|----|----|--------|------|------|----|----|----|------|-----|
| BW1 | 1 | 25 | 18 | 15 | 4 | n.d. | 1 | 27 | 6 | 4 | 11.2 | 4.3 |
| BW2 | 1 | 21 | 24 | 27 | traces | n.d. | n.d. | 23 | 3 | 2 | 11.5 | 6.2 |
| BW1 | 7 | 26 | 15 | 12 | 4 | n.d. | 1 | 24 | 8 | 10 | 13.6 | 5.8 |
| BW2 | 7 | 21 | 22 | 25 | traces | n.d. | n.d. | 18 | 4 | 10 | 14.8 | 6.3 |
| BW1 | 28 | 30 | 14 | 10 | 4 | n.d. | 4 | 16 | 10 | 12 | 12.6 | 4.2 |
| BW2 | 28 | 25 | 15 | 20 | 5 | n.d. | 5 | 11 | 7 | 12 | 11.9 | 5.7 |
| BW1 | 90 | 29 | 12 | 8 | 4 | 2 | 6 | 13 | 11 | 15 | 13.4 | 4.2 |
| BW2 | 90 | 30 | 11 | 7 | 5 | 1 | 6 | 15 | 13 | 12 | 14.8 | 5.6 |
| BW1 | 180 | 30 | 9 | 4 | 7 | 2 | 10 | 12 | 10 | 16 | 15.8 | 3.9 |
| BW2 | 180 | 29 | 8 | 4 | 7 | 3 | 10 | 14 | 9 | 16 | 16.3 | 6.2 |
| BW1 | 360 | 30 | 8 | 4 | 10 | 2 | 11 | 9 | 6 | 18 | 14.3 | 4.8 |
| BW2 | 360 | 29 | 8 | 3 | 11 | 3 | 12 | 7 | 5 | 22 | 15.0 | 5.3 |

Stratlingite ($2CaO \cdot Al_2O_3 \cdot SiO_2 \cdot 8H_2O$) is a hydrated calcium aluminosilicate that is related to the hydration reaction of aluminum-rich cements. Stratlingite was observed at 12.61 Å (7.0 2θ), 6.28 Å (14.08 2θ), 4.15 Å (21.38 2θ), and 2.87 Å (31.14 2θ) in reaction times of

more than 90 days and in a small concentration (2%) that stabilized over time. $C_4AH_{13}$ was identified at 7.9 Å (11.18 2θ), 2.88 Å (31.02 2θ), and 2.86 Å (31.24 2θ) at all ages, although in increasing amounts as the reaction time elapsed. Moreover, the LDH compounds were identified at 7.59 Å and 7.41 Å for the (006) XRD plane and at 3.78 Å for (003) XRD plane at all ages, increasing their concentrations with reaction time.

All of these phases were detected together with portlandite, whose concentrations decreased with age and with calcite, which remained almost constant. Portlandite and calcium hydroxide come from the reagents of the pozzolanic reaction that decomposed into CaO due to the handling of the materials and their reaction with atmospheric $CO_2$, after their carbonation process. As the time went by, the amorphous material increased, reaching a concentration close to 20% one year after the reaction. The new phases appeared and increased their concentrations at the expense of the initial feldspars and phyllosilicates that decreased in quantity (Table 4).

The SEM/EDX images presented in Figure 3 were used to study the morphology of the different species outlined in this article. The phases that formed as the reaction progressed were deposited on pre-existing substrates, generating aggregates of variable composition that, once analyzed, could be attributed to one species or another.

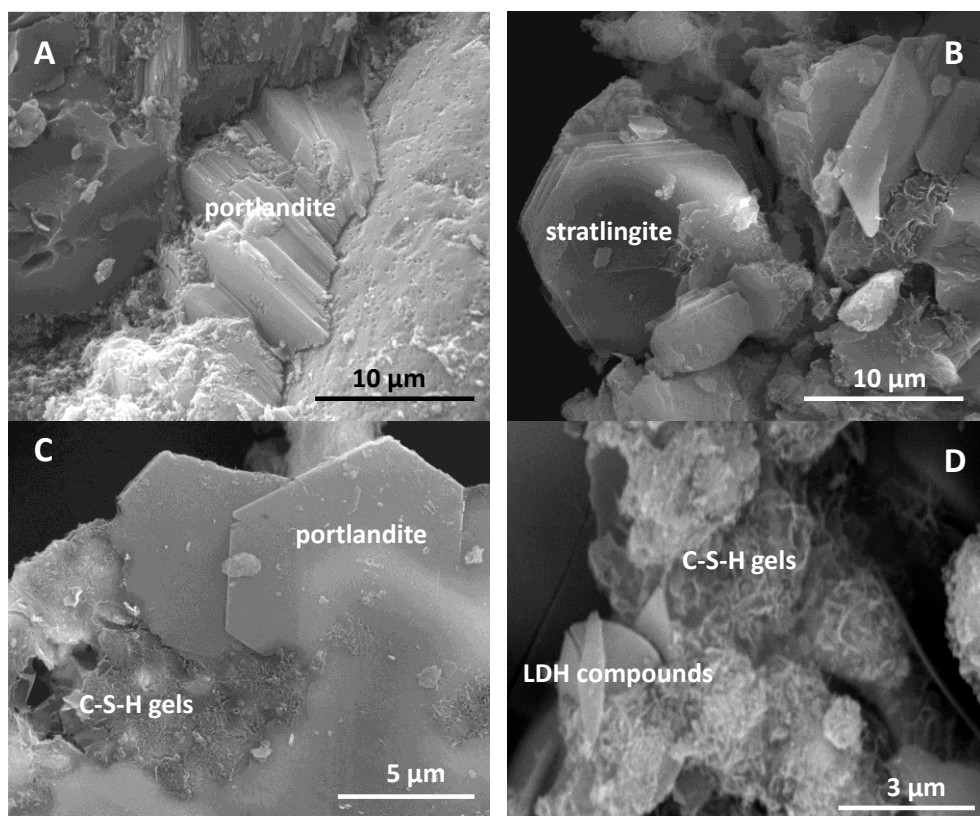

**Figure 3.** (**A**) portlandite crystals at a reaction time of 7 days; (**B**) stratlingite crystals at a reaction time of 90 days; (**C**) portlandite crystals and C-S-H gels at 180 days; (**D**) LDH compounds and C-S-H gels at 360 days.

Portlandite (Figure 3A) was recognized at all ages. In turn, stratlingite, according to the chemical analysis (Table 5), was identified as hexagonal lamellae (Figure 3B) intercalated with C-S-H gels and LDH-type compounds (Figure 3C,D). The LDH-type compounds, also of a hexagonal morphology, were identified by their proportion of $SiO_2$, $Al_2O_3$, and CaO [25] in the same way as portlandite and hydrated calcium aluminate.

**Table 5.** EDX analysis of BW1 sample (n.d. = not detected).

| Oxides (%) | C-S-H Gels | LDH | Stratlingite | Portlandite | Aluminate |
|---|---|---|---|---|---|
| $Al_2O_3$ | $18.53 \pm 0.43$ | $24.59 \pm 1.15$ | $20.72 \pm 0.54$ | n.d. | $24.32 \pm 0.41$ |
| $SiO_2$ | $26.59 \pm 1.18$ | $5129 \pm 1.12$ | $15.56 \pm 0.81$ | n.d. | $14.50 \pm 1.22$ |
| $CaO$ | $44.53 \pm 1.59$ | $32.20 \pm 1.15$ | $63.77 \pm 2.36$ | 100 | $61.18 \pm 2.06$ |
| $CaO/Al_2O_3$ | 2.42 | 1.18 | 3.07 | - | 2.51 |
| $CaO/SiO_2$ | 1.58 | 0.82 | 4.09 | - | 4.21 |
| $SiO_2/Al_2O_3$ | 1.47 | 1.39 | 0.75 | - | 0.59 |

The characterization of the solid products of the pozzolanic reaction indicated minimal crystallization in the C-S-H gels (which was not identified using XRD analyses) that comprised both the gels that have alumina and those that do not have in their structure [26]. Therefore, given the limited literature available on the pozzolanic reaction that includes the specific analysis of aqueous solutions, speciation analyses and geochemical modeling of the resulting aqueous phases were carried out after treating the waste with a saturated lime solution for 1, 7, 28, 90, and 360 days to predict the most stable associations resulting from the reaction throughout each reaction time under study.

3.3.2. Liquid Phase

The evolution of the chemical composition in the aqueous phase in contact with the pozzolan was determined by the reactions within the system. The variation in the concentrations of most elements in solution shed light on the alteration and dissolution processes of less stable mineral phases, the precipitation of new phases, and the cationic exchange reactions produced in the pozzolans.

The consumption of $Ca^{2+}_{(aq)}$ in the solution is related to the pozzolanic reaction, where the amorphous phases react with the $Ca^{2+}$ ions of the aqueous phase that are available in the alkaline medium, mainly to form gels with and without aluminum. In these type of reactions, the gels are found in metastable phases that evolve easily. In some cases, they act as precursors for the formation of other stable phases, such as stratlingite and zeolites [27].

The pH reflects the concentration of $Ca^{2+}(aq)$ in the solution. As the reaction progressed, the pH decreased and the system reached a steady state in a variable time period approaching 28 days (limiting age specified in the relevant cement standards for similar circumstances). The $Ca^{2+}(aq)$ levels fell as the solution decreased, first rapidly decreasing until 28 days and then stabilizing until one year after the reaction. This circumstance was associated with the incorporation of the calcium ions from the pozzolanic reaction to the amorphous phases, such as LDH-type compounds and C-S-H gels, at the beginning of the reaction. Over time, the metastable phases reorganize, and then the lime is consumed in the solution. The described reaction was controlled through pH, which changed at a critical age of 28 days. The concentration of aqueous silica (Figure 4) was linked to the aluminosilicate phases. In this way, a slight increase in this ion (Si) in solution was observed until 28 days of the reaction and continued to be in a steady state condition at similar concentrations, which is related to its assimilation by LDH-type compounds and C-S-H gels [28] in amounts close to 0.1 mM. The presence of $C_4AH_{13}$ was attributed to the supersaturation of calcium hydroxide in the aqueous phase and due to the low quantity of metakaolinite. The composition of the solution with high concentrations of $Ca^{2+}$ and $OH^-$, due to the dissolution of $Ca(OH)_2$, produced $C_4AH_{13}$ as a precipitate. In this study, the precipitation of $C_4AH_{13}$ occurred on day 1 and day 7 of the pozzolanic reaction.

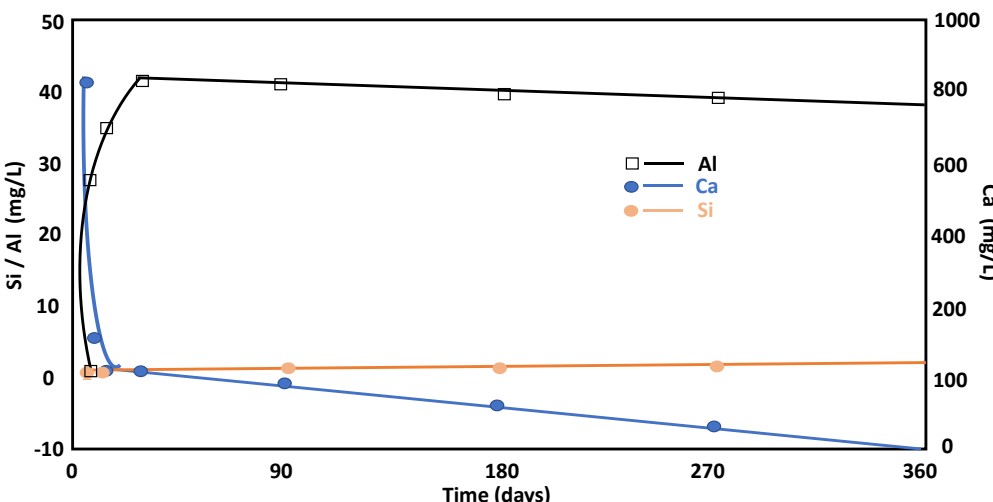

**Figure 4.** Evolution of Si, Ca, and Al concentrations throughout the pozzolanic reaction versus time.

Aqueous aluminum (Figure 4) increased in the solution during the first 7 days of the reaction and remained almost constant the rest of the time within the concentration range between 1.3 and 1.5 mM. This is a behavior that is related with the degradation of the initial minerals (feldspars and micas) and to the fact that aqueous aluminum (dominated by the $Al(OH)_4^-$ species in the alkaline medium) can only be incorporated into neoformed minerals.

The total concentration of dissolved silica in equilibrium with quartz or amorphous silica will raise the pH level (pH > 10). The total concentration of dissolved silica will be the sum of the ionized and non-ionized species $H_4SiO_4$, $H_3SiO_4$, and $H_2SiO_24$ [29]. The Ca concentrations decreased during the considered reaction time, and the Al concentration was reduced after 28 days due to incorporation of this ion in the main phase (stratlingite and $C_4AH_{13}$). The rapid decrease in the concentration of $Ca^{2+}$ can be explained by the incorporation of this ion into the structures of metastable phases, such as C-S-H gels and LDH-type compounds, which occurred during short reaction times. Over time, the metastable structures reorganized and evolved into stable phases. In these mechanisms, the incorporation of calcium ions into the stable phases is lower, which would explain the slower reduction of $Ca^{2+}$ in the dissolution process. On the contrary, an increase in Na and K concentrations in the aqueous solution was observed with time after the pozzolanic reaction. This constant increase was related to a continuous dissolution of the phyllosilicates in the pozzolanic reaction during the entire reaction time studied.

The aqueous concentrations of $Na^+$ and $K^+$ increased over time, which is related to the dissolution of feldspars and phyllosilicates from the raw material since they are the only mineral phases that contained these ions in the initial ballast waste (Figure 5). The increase in aqueous $K^+$ indicated its replacement by $Ca^{2+}$ and the non-precipitation of secondary forms with potassium throughout the reaction. The evolution of these two ions in solution (Ca and K), until day 28, indicated that continuous dissolution of the phyllosilicates in the alkaline medium occurred and that non-precipitation of new phases occurred in tandem. A large portion of the ionic salts released in the cement mixtures were probably incorporated into the solid cement.

Aqueous $Mg^{2+}$ does not provide relevant information because its concentration is minimal (Figure 5). Therefore, the values suggested the incorporation of Mg at the beginning of the pozzolanic reaction in the C-S-H gels, which coincides with its mineralogical composition. Finally, Mg concentrations were very low throughout the reaction (Figure 5). The results are like those obtained through Argentine kaolinitic clays, for example [30].

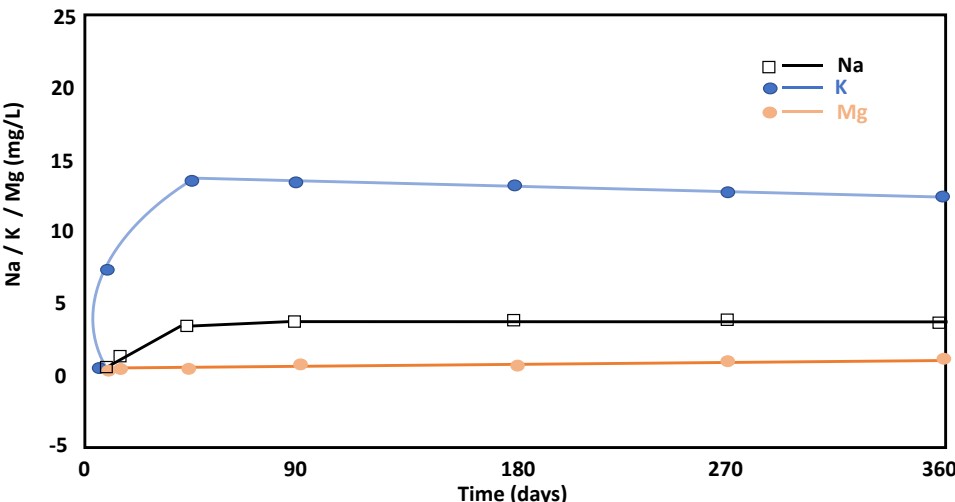

**Figure 5.** Evolution of Na, K, and Mg concentrations throughout the pozzolanic reaction versus time.

The saturation index for portlandite varied from a value of zero at the beginning of the pozzolanic reaction to various negative values, given that the mineral was in equilibrium, both in the initial saturated solution and during the pozzolanic reaction (Figure 6). The LDH-type compounds and stratlingite were more likely to precipitate. The compounds were more stable throughout the reaction (1, 7, and 28 days), with low stability at 90 days, while stratlingite increased its stability. The high saturation levels of $C_4AH_{13}$ meant that it could not precipitate under these conditions, as suggested by the high Al concentrations in the solution. However, the C-S-H gels were more stable, as evidenced by Ca/Si ratio increasing on day 1, and their stability decreased as the Ca/Si ratio increased over time due to the remaining $Ca^{2+}$ concentration in the solution (Figure 6). At early stages with high aqueous $Ca^{2+}$, C-S-H gels with higher Ca/Si ratios were favored in formation from a thermodynamic point of view, and the gels with lower Ca/Si ratios formed more quickly due to kinetic considerations. As the reaction continued, the availability of $Ca^{2+}$ in the aqueous system was reduced and C-S-H gels with lower Ca/Si ratios were formed.

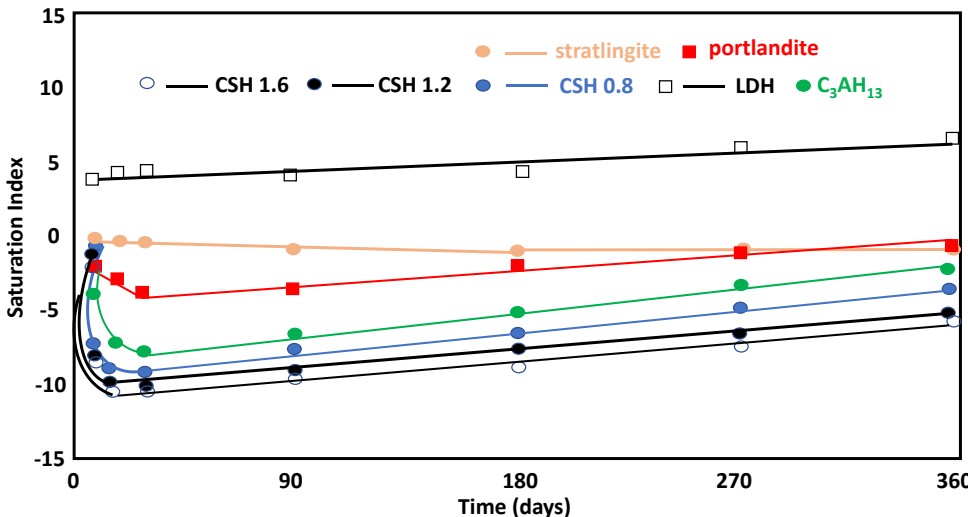

**Figure 6.** Evolution of the saturation index and the reaction time of neoformation phases.

Stratlingite and LDH-type compounds, both with positive saturation indices, are phases with a tendency to precipitate, coinciding with the experimental observations. It is believed that C-S-H gels with higher Ca/Si ratios are thermodynamically more stable in

the short term (1 day) and are more unstable in the long term (7, 28, 90, 180, and 360 days), which was justified by the concentration of dissolved $Ca^{2+}$(aq).

Thus, in the short term, when the concentration of $Ca^{2+}$(aq) is high, the formation of C-S-H gels with high Ca/Si ratios is favored. As the reaction continues, the availability of $Ca^{2+}$ within the aqueous medium is reduced and, therefore, the formation of C-S-H gels with lower Ca/Si ratios can be observed in the thermodynamic analysis. However, if kinetic considerations are considered, C-S-H gels with lower Ca/Si ratios could form more rapidly, even to the point where $Ca^{2+}$ ions could be incorporated within a previously formed C-S-H gel.

The standard thermodynamic properties required to calculate the equilibrium constants for each reaction ($\Delta G0$ [J/mol], $\Delta H0$ [J/mol], S0 [J/mol K], and V0 [$cm^3$/mol]) were selected based on "cemdata2007" data [30]. In Figure 7, the stability fields of these phases are represented as a function of silica activity and the coefficient between the activities of calcium and the proton in solution. The activities of the aqueous species, previously calculated with PHREEQC [31], were clearly located in the stability field of the C-S-H gel, approaching, as time passes, the stability zone of the stratlingite. This agrees with the generation of C-S-H gels throughout and agrees with the appearance of stratlingite at the later stages.

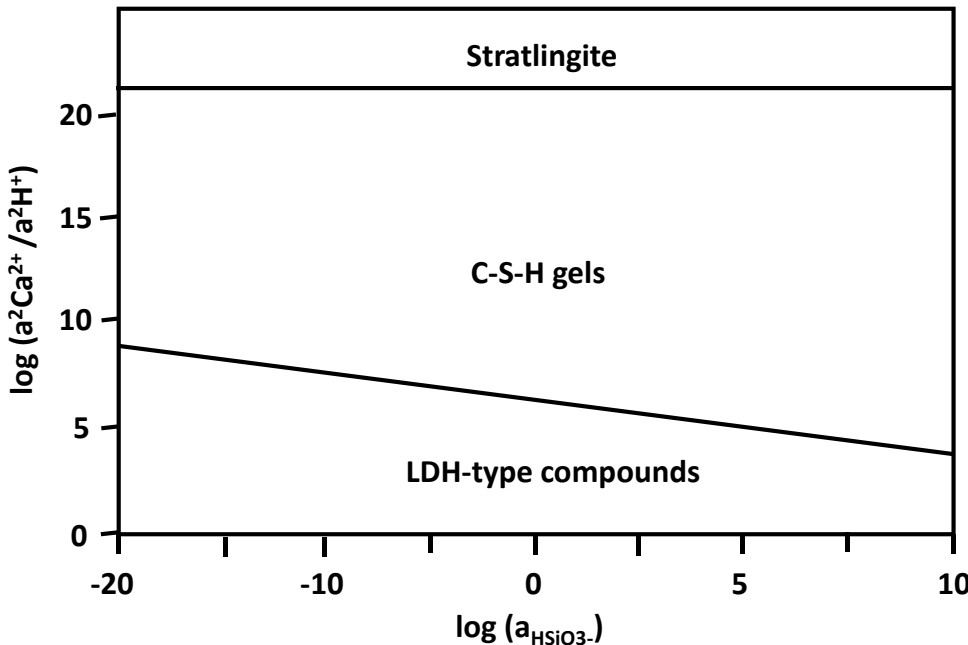

**Figure 7.** Diagram of the stability fields of the LDH-type compound/stratlingite/C-S-H gel systems.

Although no precipitation of C-S-H gels was predicted in the thermodynamic analysis described, the three phases considered are close to equilibrium; thus, their precipitation is attributed to kinetic factors.

According to the temporal evolution of the saturation indices obtained after the chemical speciation of the experimental phases, the most stable association in the system was defined, considering the neoformed phases of the pozzolanic reaction. As a criterion, it was established that the most stable phases—LDH-type compounds ($Si_2Al_2(CO_3)(OH)_{12}$) and stratlingite ($Ca_2Al_2(SiO_2)(OH)_{10}$)—presented positive saturation indices [32], i.e., they were supersaturated in the aqueous phases that were analyzed [33]. In the case of the C-S-H gels, those that constituted the amorphous phase quantified with XRD analysis and that were close to equilibrium were considered more stable. Despite the negative sign, the probability of gel formation with Ca/Si ratios equal to 0.83 of the tobermorite type and a composition of $CA_{0.83}SiO_2(OH)_{1.66}$ is greater.

## 4. Conclusions

1.  The conclusions drawn from the experimental results were as follows. The ballast waste is made up of silicate minerals, such as phyllosilicates 2:1 (biotite), quartz, potassium-rich feldspars in (orthoclase), and plagioclase (labradorite).
2.  The reactions products were formed from the original materials, except for quartz, which is very stable.
3.  Stratlingite, LDH-type compounds, and $C_4AH_{13}$ are the main crystalline products of the pozzolanic reaction detected using XRD analysis throughout the reaction.

The precipitation of stratlingite during the pozzolanic reaction was calculated with the PHREEQC method. The results indicated that stratlingite is one of the main phases produced during the pozzolanic reaction, which agrees with the experimental XRD and SEM data. The use of ballast waste as a pozzolanic addition is a reasoned and technically viable scientific option that eliminates large quantities of waste, making it applicable for use in the manufacture of cement.

**Author Contributions:** Conceptualization, R.G.-G.; methodology, S.Y.-G.; software, S.Y.-G.; formal analysis, R.G.-G.; writing-original draft preparation, S.Y.-G.; visualization and supervision, R.G.-G. All authors have read and agreed to the published version of the manuscript.

**Funding:** The authors received no financial support for the research, authorship, and/or publication of this article.

**Data Availability Statement:** No new data were created or analyzed in this study. Data sharing is not applicable to this article.

**Conflicts of Interest:** The authors declare no conflicts of interest.

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
