# Peer review of "Construction and Demolition Waste Ballast as a Pozzolanic Addition in Binary Cements: Characterization and Thermodynamic Stability"

_minerals, doi:10.3390/min14040402_

Round 1
Reviewer 1 Report
Comments and Suggestions for Authors
This study uses deteriorating railway ballast, which is quartz gravel. This substance was added to mixes to replace part of the cement, creating a high-quality cement composite. For one year, the waste/lime system in the solid phase and water system were studied, observing the formation of stratlingite, LDH compounds, hydrated tetracalcium aluminate, etc.
The work is well written, the methods applied are consistent with the objectives, and the results are well discussed
Before the article is accepted, my suggestions are as follows:
1) Report whether this material has already been used as a pozzolanic material. If not, emphasize the innovation of the work. If yes, insert the literature review in this regard
2) Insert indications and a slightly more detailed description of figure 3 of the SEM. Because images must be accessible to the reader
3) Table 5 shows the composition of oxides on the surfaces and refers to the EDX results. In EDX, the composition given is usually for each element. I would like the authors to explain how they created this table
4) Figure 6 shows the saturation index, I would like more details on how this index was obtained
Author Response
We appreciate the interest in improving the work and the suggestions made by the reviewer, as well as the editor. A complete review of the language and coincidences with other works has been made, trying to avoid the same terms. The corrections have been marked in red.
Reviewer 1
This study uses deteriorating railway ballast, which is quartz gravel. This substance was added to mixes to replace part of the cement, creating a high-quality cement composite. For one year, the waste/lime system in the solid phase and water system were studied, observing the formation of stratlingite, LDH compounds, hydrated tetracalcium aluminate, etc.
The work is well written, the methods applied are consistent with the objectives, and the results are well discussed
Before the article is accepted, my suggestions are as follows:
- Report whether this material has already been used as a pozzolanic material. If not, emphasize the innovation of the work. If yes, insert the literature review in this regard
Ballast waste has not been used as pozzolanic material yet. Its use is proposed above all for slab track, since another type of utility in the railway sector is the manufacture of sleepers and this use depends on the manufacturers that admit the waste that is being studied in their compositions.
- Insert indications and a slightly more detailed description of figure 3 of the SEM. Because images must be accessible to the reader
Figure 3 brings together the evolution of the different phases that have appeared over time in replacement cements. It corroborates what was detected by XRD, only manifested in a more visual way. The figure has been changed since the identification of the phases mentioned in the text has been included on the photographs.
- Table 5 shows the composition of oxides on the surfaces and refers to the EDX results. In EDX, the composition given is usually for each element. I would like the authors to explain how they created this table
Indeed EDX provides data with values for each chemical element. A simple rule of three allows us to extrapolate these data from elements to oxides, for example for aluminum it would be enough to multiply the quantity obtained in the Al reading in EDX by the quotient between the Molecular Weight of alumina and the molecular weight of aluminum.
4) Figure 6 shows the saturation index, I would like more details on how this index was obtained
The Saturation Index (SI) is obtained from the analysis of the ions in solution analyzed at each of the ages and by the PHREEQC method. In each case a positive SI suggests the precipitation of the corresponding mineral, while if it is negative it is a phase that is in solution. Finally, if the value is zero it means that the mineral is in chemical equilibrium with the solution.

Reviewer 2 Report
Comments and Suggestions for Authors
good scientific research, but it is difficult to see any practical application to civil engineering; the pozzolanic reactivity is clear but what are the best application in cement and concrete industries: addition for cement production ? substitution in concrete mix of what?
Have you any ideas of percentage of substitution of clay in clinker production or of addition in concrete mix?
Author Response
We appreciate the interest in improving the work and the suggestions made by the reviewer, as well as the editor. A complete review of the language and coincidences with other works has been made, trying to avoid the same terms. The corrections have been marked in red.
Reviewer 2.
Good scientific research, but it is difficult to see any practical application to civil engineering; the pozzolanic reactivity is clear but what are the best application in cement and concrete industries: addition for cement production ? substitution in concrete mix of what?
The application in the cement industry is to use this waste (avoiding unnecessary consumption of raw materials and following the Circular Economy) that until now has been used as sterile or has been sent to landfill as a secondary material in the manufacture of cement, as an addition pozzolanic. It could be used in the railway sector to manufacture concrete sleepers or to make slab track. At the moment it is a little-disclosed investigation.
Have you any ideas of percentage of substitution of clay in clinker production or of addition in concrete mix?
In a previous work (Yagüe García, S.; González Gaya, C. Reusing Discarded Ballast Waste in Ecological Cements. Materials 2019, 12, 3887), it has been seen that the substitution can reach 20% and the properties are maintained.

Reviewer 3 Report
Comments and Suggestions for Authors
The research investigates the use of railway ballast waste material as pozzolan in cement composites. The pozzolanic activity was determined using a saturated lime solution and chemical and mineralogical investigation. Rock minerals used in ballast contain oxides that could possess pozzolanic reactions, favoring cement hardening and soluble compound transformation in HCP. The subject area previously was covered by waste ballast investigation through road building standards, components of cementitious materials as aggregate, and other rock mineral reactivity in cementitious materials. This paper addresses a novel approach to the use of waste ballast as supplementary cementitious material, and the reactivity is tested in saturated lime solution.
Methodology.
This is the subject which needs improvements. It is not clear how the waste ballast was prepared for the tests. Please describe the treatment of the waste material, both C and Cu. Materials: Please indicate the lime used in the experiments (mentioned in the abstract). C and Cu are unsuitable codes for material abbreviations as they are more known as chemical elements. I suggest replacing this nomenclature with Hornfel rock HOR, CAC- Canteras Cuadrado, or similar.
Conclusions are consistent and are based on the research and obtained results. The treatment of the waste ballast could be mentioned, and the amount of waste ballast that could be effective in cementitious materials could be proposed.
The section introduction is clearly written and covers the scope of the work with the latest literature review. References are appropriate and provide background information about the research.
Additional comments: Abstract: Justification is needed on how the gravels can substitute cement. Describe the treatment of the gravel. Avoid using abbreviations in the abstract (without explanation) – LDH, PHREEQC.
Author Response
We appreciate the interest in improving the work and the suggestions made by the reviewer, as well as the editor. A complete review of the language and coincidences with other works has been made, trying to avoid the same terms. The corrections have been marked in red.
Reviewer 3
The research investigates the use of railway ballast waste material as pozzolan in cement composites. The pozzolanic activity was determined using a saturated lime solution and chemical and mineralogical investigation. Rock minerals used in ballast contain oxides that could possess pozzolanic reactions, favoring cement hardening and soluble compound transformation in HCP. The subject area previously was covered by waste ballast investigation through road building standards, components of cementitious materials as aggregate, and other rock mineral reactivity in cementitious materials. This paper addresses a novel approach to the use of waste ballast as supplementary cementitious material, and the reactivity is tested in saturated lime solution.
Methodology. This is the subject which needs improvements. It is not clear how the waste ballast was prepared for the tests. Please describe the treatment of the waste material, both C and Cu.
The simulation has been carried out by collecting rock fragments from the aforementioned quarries and subjecting them to the Los Angeles wear test. In this test, the fragments are rounded and release a fine material that is equivalent to what is actually produced on the road and that is the fine-grained product that is used as SMC.
Materials: Please indicate the lime used in the experiments (mentioned in the abstract).
Calcium hydroxide used in the pozzolanic reaction is the commercial powder product, PANREAC brand, with an extra pure purity index Ph Eur USP.
C and Cu are unsuitable codes for material abbreviations as they are more known as chemical elements. I suggest replacing this nomenclature with Hornfel rock HOR, CAC- Canteras Cuadrado, or similar.
Thank you very much for your appreciation, it may be misleading. The names C have been replaced by BW1 and Cu by BW2.
Conclusions are consistent and are based on the research and obtained results. The treatment of the waste ballast could be mentioned, and the amount of waste ballast that could be effective in cementitious materials could be proposed.
In a previous work (Yagüe García, S.; González Gaya, C. Reusing Discarded Ballast Waste in Ecological Cements. Materials 2019, 12, 3887), it has been seen that the substitution can reach 20% and the properties are maintained.
The section introduction is clearly written and covers the scope of the work with the latest literature review. References are appropriate and provide background information about the research.
Additional comments: Abstract: Justification is needed on how the gravels can substitute cement. Describe the treatment of the gravel. Avoid using abbreviations in the abstract (without explanation) – LDH, PHREEQC.
Abbreviations have been removed from the abstract.

Reviewer 4 Report
Comments and Suggestions for Authors
See attached file.

Author Response
We appreciate the interest in improving the work and the suggestions made by the reviewer, as well as the editor. A complete review of the language and coincidences with other works has been made, trying to avoid the same terms. The corrections have been marked in red.
Reviewer 4
Commend on minerals-2933663
It is lack of the existing status of used waste ballast.
What is their particle size distribution? Are they coarse aggregate or powder?
The granulometric distribution curves are similar for the two samples studied and offer two maxima, one around 6 μm for both and a second, at 9.5 μm, for CU and 15 μm for C.
What is the mix proportion of samples?
In a previous work (Yagüe García, S.; González Gaya, C. Reusing Discarded Ballast Waste in Ecological Cements. Materials 2019, 12, 3887), it has been seen that the substitution can reach 20% and the properties are maintained.
Based on Table 2 it can be found that C and Cu samples contain only little amorphous phase. Therefore, these rock cannot have high pozzolanic reactivity. They cannot used as supplemental cementitious material. It is very important to determine the effect of used waste ballast on the mechanical properties of binder containing some percentage of waste ballast. It is lack of this work in this paper
The results of using portions of waste as a substitute for cement fractions and their applications are satisfactory and have already been published in Yagüe García, S.; Gonzalez Gaya, C. Durability analysis of pozzolanic cements containing recycled track ballast: Sustainability under extreme environmental conditions. Constr Build Mater. 2020, 242, 117999.

Round 2
Reviewer 3 Report
Comments and Suggestions for Authors
Authors have addressed all the concerns and recommendations given by the reviewer.
Small correction which should be covered about the result representation:
In the text it is said that XRD are identified at different reaction times (1, 7, 28, 90, 180, and 360 days) (Figure 2). While in fig.2 no pattern for 90th day is represented.
Author Response
Sorry for the forgetfulness and thank you very much for your supervision T
he pattern for 90 days has been added to Figure 2